# A Literature Review on Psychosocial Support for Disaster Responders: Qualitative Synthesis with Recommended Actions for Protecting and Promoting the Mental Health of Responders

**DOI:** 10.3390/ijerph17062011

**Published:** 2020-03-18

**Authors:** Maki Umeda, Rie Chiba, Mie Sasaki, Eni Nuraini Agustini, Sonoe Mashino

**Affiliations:** 1Research Institute of Nursing Care for People and Community, University of Hyogo, 13-71 Kitaoji-cho, Akashi, Hyogo 673-8588, Japan; mie_sasaki@cnas.u-hyogo.ac.jp (M.S.); sonoe_mashino@cnas.u-hyogo.ac.jp (S.M.); 2Graduate School of Health Sciences, Kobe University, 7-10-2 Tomogaoka, Suma-ku, Kobe, Hyogo 654-0142, Japan; crie-tky@umin.ac.jp; 3Graduate School of Nursing Art and Science, University of Hyogo, 13-71 Kitaoji-cho, Akashi, Hyogo 673-8588, Japan; eni.nuraini@uinjkt.ac.id; 4School of Nursing, Faculty of Health Sciences, Syarif Hidayatullah State Islamic University, Jl. Kertamukti No.5 Cireundeu CiputatTangerang Selatan, Banten 15419, Indonesia

**Keywords:** disaster responders, support, psychosocial, risk management

## Abstract

Little scientific evidence exists on ways to decrease the psychological stress experienced by disaster responders, or how to maintain and improve their mental health. In an effort to grasp the current state of research, we examined research papers, agency reports, the manuals of aid organisations, and educational materials, in both English and Japanese. Using MEDLINE, Ichushi-Web (Japanese search engine), Google Scholar, websites of the United Nations agencies, and the database of the Grants System for Japan’s Ministry of Health, Labour, and Welfare, 71 pertinent materials were identified, 49 of which were analysed. As a result, 55 actions were extracted that could potentially protect and improve the mental health of disaster responders, leading to specific recommendations. These include (1) during the pre-activity phase, enabling responders to anticipate stressful situations at a disaster site and preparing them to monitor their stress level; (2) during the activity phase, engaging in preventive measures against on-site stress; (3) using external professional support when the level of stress is excessive; and (4) after the disaster response, getting back to routines, sharing of experiences, and long-term follow-up. Our results highlighted the need to offer psychological support to disaster responders throughout the various phases of their duties.

## 1. Introduction

Natural disasters threaten every aspect of people’s lives and are a significant burden on the health of those affected [1,2]. The alleviation of the health impacts resulting from disasters is of great concern to all countries, as the number and scope of disasters have been increasing worldwide due to global trends in urbanisation, environmental degradation, and climate change [3]. 

In the acute phase of a disaster response, treating physical health problems has often been prioritised. However, over the past two decades, the psychological impact of disasters has come to be a major focus of disaster health management [4,5,6]. The morbidity of psychiatric disorders after disasters has been reported to be as high as 60% [7], with an increased risk for anxiety, depression, stress-related disorders, and alcohol and substance abuse [8]. Deteriorating psychological health substantially decreases quality of life, and negatively affects physical and social functioning [9]. These facts highlight the importance of enhancing psychological health as critical to alleviating the health impact of disasters [10,11]. 

Disaster responders, such as health professionals, relief workers, public-service providers, and volunteer workers in disaster-affected areas, are at high risk for extreme stress, as are disaster survivors. Being a disaster responder involves exposure to traumatic events, a high level of work demands with limited resources, working with highly stressed populations in critical moments, and separation from home and family [4]. In addition, earlier studies have demonstrated that disaster responders generally felt unprepared and were not confident they would be able to effectively support others [12,13], which can result in psychological exhaustion and burn out. 

In spite of the high likelihood of disaster responders experiencing mental health problems, little research has been conducted on ways to decrease their psychological stress and maintain or improve their mental health [14]. Existing studies suggest that programs providing knowledge about stress and stress management could improve the self-esteem of disaster responders, facilitate their self-care, and motivate them to engage in self-directive learning regarding their duties [15,16,17,18]. Programs aiming to alleviate existing psychological symptoms have also been found to be useful by some researchers [19]. On the other hand, a few studies did not find such programs effective [20,21]. Still other programs targeting disaster responders’ mental health lack empirical evidence to support their effectiveness [22,23].

The present review was part of a project that aimed to develop a psychosocial support guide for disaster responders that could be used in a global setting. The purpose of this review was to identify the types of psychosocial support considered appropriate for disaster responders as a preliminary item pool for guide development. To achieve this aim we reviewed the non-academic literature, such as guides, manuals, and educational materials, to identify field-based knowledge and practices. Although the information in this review may not necessarily be tested scientifically, it does provide a comprehensive picture of field experiences that could be further scrutinised by scientific measures. Academic articles examining the effectiveness of psychosocial support specific to disaster responders were also reviewed as a primary source of evidence. Materials in English and Japanese do not cover all the generated information in other parts of the world, but it is anticipated that reports from Japan, which is one of the most disaster-prone countries and thus has a rich experience in disaster responses [24], could provide a solid knowledge base for exploring this issue. 

In the following sections, we describe the procedure of our review and our analytical framework. Accordingly, our findings are presented using an analytical framework that groups the identified actions by their goals. To enhance their applicability to the field, the identified actions were further grouped by disaster response phase and actors. Lastly, we discuss the characteristics of these actions, and challenges in their implementation.

## 2. Material and Methods 

### 2.1. Review Authors

All five authors conducted the literature search, and four of them analysed the data. All authors were from a nursing research institute specializing in disaster health management. Two of the authors are specialised in psychiatric nursing; one in nursing management and disaster nursing, one in acute care management, and the other in public health and epidemiology. All authors were project members for the development of a guide on psychosocial support for disaster responders, and all have field or research experience in disaster-affected areas.

### 2.2. Search Strategy

The search was conducted by all authors on MEDLINE (OVID), Ichushi-Web (Japanese search engine), Google Scholar, websites of United Nations (UN) agencies, and the database of the Grants System for Japan’s Ministry of Health, Labour, and Welfare. The employed key search terms were “disaster”; “providers or responder”; “mental or psychological or psychosocial”; “support, education, or intervention”; and “critical incidence stress”. The reference lists of relevant studies and reviews were also checked, and as a result, individual book chapters and educational brochures were included in our examination. Language was limited to English and Japanese. Below is a flow chart showing the study selection process (Figure 1). 

### 2.3. Inclusion Criteria

All types of materials, such as guidelines, manuals, educational materials, and research reports, were reviewed if they provided detailed information about actual or recommended psychosocial support for disaster responders. The initial focus of this review was psychosocial support for disaster responders who had not received formal training to respond to a natural disaster. Materials developed for firefighters and army personnel were not included because the support needs of these professionals would most likely differ from our target responders, owing to the difference in preparedness for critical-disaster-related incidents. On the other hand, we did not exclude materials with a broader scope than natural disasters when they provided applicable information to disaster responders.

### 2.4. Term Definitions

Psychological support is a composite term that was defined as any type of internal and external support that aims to protect or promote psychosocial well-being, prevent mental disorders, and facilitate treatment if needed [25]. 

### 2.5. Analysis

The analysis procedure was conducted by four review authors. At the beginning of the analysis, the authors discussed the analytical framework. With the aim of developing a guide applicable to broad societal contexts, goals were developed which were focused on enhancing field applicability by allowing variations in local contexts. Before setting the goals, we derived some actions from identified materials, and sorted these by disaster-response phase. Next, we categorised the actions into a group that aimed at the same goal (these were labelled “Goals”). Here, we explain in greater detail the procedures in each step by asking research questions. 

The first question was “What should be done to protect or improve the mental health of responders?” To answer this question, the recommended actions for protecting and improving the mental health of disaster responders were extracted from the materials, and each action was separately recorded on a Post-It note. Upon reviewing these notes, it became apparent that there were two groups initiating these actions. The first was organisations that dispatch responders to disaster-affected areas or co-ordinate responders on site. The second was made up of disaster responders themselves, who were expected to maintain and enhance their own mental health. 

The second question was “When should these actions be taken?” To answer this question, the various Post-It notes were grouped together on the basis of which activity phase the action was part of: the pre, during, or after phase of the disaster response. At this stage, similar types of data were gathered as a set of cards, and examined on the basis of whether they should be grouped as a single unit or independent sets. Then, a label was selected for a set of cards on the basis of reviewer consensus. 

Once all information was classified by phase and actor, the final question was “For what reason should these actions be taken?” In exploring the goals of these actions, we focused on the stress and coping theory of Lazarus that conceptualises coping as a process of interpreting the cause of psychological stress (stressors), evaluating coping options, taking actions to reduce stress, and reappraising the coping process [26]. The three identified goals were (1) understanding stressors and making them manageable, (2) reducing stressors and preventing chronically stressful situations, and (3) responding to crises for those whose level of stress was overwhelming and something that could not be handled with normal coping strategies. Each action on the cards was classified under one of these goals to demonstrate the expected achievement of these actions.

## 3. Results

The search identified 71 materials and, eventually, 49 were used for analysis (Table 1, Table 2 and Table 3). Fifty-five actions that potentially protect and improve the mental health of disaster responders were extracted. Each action is explained below on the basis of its goal.

### 3.1. Goal 1: Understanding Stressors and Making them Manageable

Seventeen actions were identified under this goal (Table 1). Responders were encouraged to gather information on their duties and the area of operation, and then assess their readiness to join the disaster response team before enrolment [27,28,29,30,31,32,33,34,35]. Organisations were recommended to train responders to monitor their own stress levels for better stress self-management during a disaster response [17,21,27,28,31,32,35,36,37,38,39,40,41,42,43,44,45,46,47]. Making a thoughtful decision on who should be a member of the response team was also considered part of the role of the organisation before enrolment [28,36,37,42,43,47].

During the activity phase, monitoring stressors and their impact was considered a significant action for protecting responders’ mental health at both the individual and the organisational level [25,27,29,30,37,38,41,45,48,49]. The re-conceptualisation of experience and feelings was specifically suggested for individuals as a way of taking an objective view of stressors and enhancing their coping ability [29,30,41,50]. 

After the activity, a reappraisal of experience and feelings during the activities was also recommended. In this post-phase, organisations were expected to provide opportunities for this reappraisal in the form of workshops, seminars, and the like [25,35,39,44,45,50,51,52,53,54,55]. Furthermore, organisations were encouraged to recognise the responders’ work as an essential contribution to the organisational goals, and to demonstrate their appreciation to members of the organisation [30,35,41,50,56]. Continuous monitoring of responders’ mental health was another action to be taken by both individuals and organisations [30,41,57,58,59,60]. 

### 3.2. Goal 2: Reducing Stressors and Preventing Chronically Stressful Situations 

Twenty-five actions were found under this goal (Table 2). In addition to maintaining physical and mental wellbeing, solving concerns at home and work before enrolment were considered necessary preparation for lightening possible stressors that might add to the burden during disaster-response activities [25,28,35]. It was also recommended that responders learn stress-management skills and make their own self-care plan so that they can effectively respond to on-site stress [28,32,40,41,50], while organisations were expected to provide the relevant training opportunities [23,25,27,28,29,35,37,38,39,43,47,48,50,60,61,62]. In addition, improving teamwork, ensuring responders have clear ideas about their duties, and making an efficient operational system prior to deployment were considered effective ways of preventing stressful situations [28,41,44,48,50].

During the activities, responders were encouraged to take care of their physical and mental health. Maintaining their routines, refraining from use of stimulants or addictive substances such as caffeine, tobacco, and alcohol, and carrying out their own self-care plans were highly recommended [24,27,28,29,30,31,32,38,41,42,45,48,53,59,63]. On the organisational side, several actions were recommended, such as managing the responders’ workload and duties, providing them with opportunities for informal communication and peer support, and holding defusing sessions [25,27,28,29,30,37,38,43,44,45,46,48,50,54,64,65,66]. 

After completing on-site duties, responders were encouraged to take time off from work in order to promote recovery from physical and psychological fatigue [27,29,40,59,67,68,69]. Making sure that responders can take this rest before coming back to routine work was recommended for organisations [24,25,39,41,50,57,70]. Changing from “disaster-response mode” to “routine mode” was seen as another effective means for maintaining the mental health of responders [30]. Offering a clear declaration of mission accomplishment and keeping responders informed about self-care were effective actions that could be taken by organisations [24,28,37,39,48,57,70]. 

Family was considered as a significant source of informal support for responders. Communication between responders and family members about their duties prior to deployment could be key to ensuring their families are supportive during and after deployment [28,32]. Organisations were also encouraged, prior to deployment, to develop a means of communicating with the responders’ family during the deployment [43,50]. Staying connected to their family during deployment was also recognised as an effective way for responders to cope with their stress [25,28]. 

### 3.3. Goal 3: Alleviating Stressful Situations and Managing Crises 

Thirteen actions were identified under this goal (Table 3). Becoming aware of one’s own indicators of extreme fatigue and crisis before enrolment could be helpful to capture signs of crisis on site in a timely manner [32]. On the part of organisations [28,36,37,43,45,48,50,56], providing responders with psychological first-aid training was highly recommended in order to enable responders to be effective supporters to their peers. Building a system that effectively responds to the traumatic experience of responders prior to deployment was also considered an organisational responsibility. 

Once stress becomes unavoidable, responders’ workload and duties should be adjusted in order to ease their physical and psychological burden [25,28,37,42,46,48,56]. If the level of stress seems overwhelming, the use of mental-health professionals was suggested, and a decision as to whether an individual responder should continue with or resign their duties should be made [24,25,28,31,37,42,46,48,56,66]. 

After the completion of activities, responders who suffered from stress need rest for a considerable amount of time [37,56,57,71]. The use of resources, including mental health professionals, was also suggested in this phase [24,48,49,53,57,58,59]. In addition, adjusting the responder’s routine work if the level of stress continued was considered part of the organisation’s role [24,25,27,28,30,36,41,48,49,59,72].

## 4. Discussion

The identified actions in this study were implemented throughout all phases of disaster-response activities. Fifty-five actions to protect and promote the mental health of disaster responders were identified from the reviewed materials, and the three following goals were derived from these actions: “understanding stressors and making them manageable’’ (Goal 1), “reducing stressors and preventing chronically stressful situations’’ (Goal 2), and “alleviating stressful situations and managing crises’’ (Goal 3). These three goals can be summarised as the following three steps: “be aware”, “prevent”, and “respond”.

Among the 55 actions, 17 fell under Goal 1, 25 under Goal 2, and 13 under Goal 3. Therefore, about 70% of the actions were preventive measures (“be aware” and “prevent”). Although responding to a stressor once it occurs is still an important countermeasure against negative impacts on metal health among disaster responders [9], understanding and reducing the occurrence of stressors and enhancing resilience can also be a major part of psychosocial support for disaster responders.

In regards to the timing of actions, most actions were to be performed before and during a disaster response, with 20 actions during the pre-period, 21 during an activity, and 14 during the post-period. Specifically, the largest number of actions for Goal 1 (“be aware”) were found in the pre-period, for Goal 3 (“respond”) in the activity period, and for Goal 2 (“prevent”) in both the pre- and during periods. These results indicate that many types of psychosocial support were expected to be performed at the earlier stages of a disaster response. This notion echoes the view that pre- and peri-disaster preparation aimed at reducing stressors and responding to stress reactions can help prevent post-disaster mental-health problems [5].

Our findings also suggest that individual actions are as important as organisational support for the mental health of disaster responders, as almost the same number of actions were identified for individuals as for organisations. Individual actions, such as monitoring one’s own stress level, managing stressors, and performing self-care, were considered to be essential, especially in disaster-affected areas, where each responder is perceived as a caregiver and works under very high-stress conditions with limited human and social resources [4]. Many organisational actions, such as “train responders to monitor their stress” (Goal 1), “improve the responders’ basic knowledge and stress-management skills” (Goal 2), and “provide information on self-care” (Goal 2), were undertaken to support such individual actions.

Another important role of organisations was to develop a system to monitor, reduce, and respond to responders’ stressors and stress reactions, including “develop an efficient operational system with a clear command chain”, “have a written and active policy for preventing and managing stress among responders”, “control the volume and content of work”, and “develop a system that responds to the traumatic experiences of responders”. Furthermore, recommendations such as “create mutually supportive teams with co-workers”, “support responders’ informal communications with their peers”, and “develop a peer support system within the team” suggest that the use of a peer support system may be a promising psychosocial approach to building a resilient response team.

On the basis of these results, the following types of psychosocial support were considered to be helpful. The first focuses on preparing disaster responders for stressor management. Psychosocial support of this kind can be offered through training and knowledge dissemination on stress management, disaster-response duties, and team-building skills [4]. The second is to build the operational system, specifically in terms of reducing stressors from daily activities. Ongoing support, both informal and formal, needs to be established on a daily basis and over the long term [37]. Such support is crucial to maximising the productivity of the response team and achieving organisational goals, as well as protecting workers from stress-related mental disorders [37,50]. The third type is crisis support and management. When there is the need for timely support, external professional support can be helpful [37]. Clear written policies and manuals for its administration are needed to make it effective [37,41,48,49].

However, recommending these actions does not necessarily mean that they were implemented in the field. For example, a study conducted in the area affected by the Great East Japan Earthquake [50] found that a significant proportion of firefighters (42%) reported that long-term leave was the most prioritised measure for treating critical stress-related incidents caused by the disaster response. On the other hand, only 2% reported that their affiliated department offered them leave after completion of their response duties. Considering that human-resource management for firefighters is often better prepared for disaster situations, the unmet needs of other types of disaster responders, especially volunteer responders, may be even larger.

One of the possible barriers causing the gap between recommendations and implementation may be conflicts between routine work and disaster response. In the case that disaster response is not part of a routine, organisations may not be prepared when their staff leave work behind to respond to a disaster. A lack of supplementary staff to make up for their work would heavily burden their colleagues and the management department. Under such conditions, psychosocial support, such as training, monitoring, and offering long-term leave, may not be readily available for responders. In addition, personal characteristics, such as “calm and emotionally collected, acts on logic over emotion, exercises emotional control and self-control”, are highly valued among disaster responders [73]. Therefore, seeking support for one’s mental health conditions can be stigmatised in the culture of responders [50]. Individual counselling for all involved workers may be an effective way of reaching out to such responders [50]. This approach could eventually promote awareness that psychological reactions caused by a disaster response can affect anyone.

This study has the following limitations. First, the reviewed materials were all written in either Japanese or English. Although the derived information from the Japanese materials grew out of extensive experience in disaster response, the contexts of the disasters and the roles of the disaster responders may differ on the basis of geographical areas and cultural regions. In addition, our review covered recommended actions written in reports, guides, manuals, and research articles. Therefore, how many of these actions were performed in the field, and whether they were effective for protecting and improving the mental health of responders, remain unclear. For example, a systematic review conducted by Guilaran et al. [74] reported that the positive impact of interpersonal support was limited in the prevention of mental health problems among disaster responders. Thus, to develop an evidence-based package of psychosocial support for disaster responders, further studies examining the feasibility and effectiveness of these actions are needed.

## 5. Conclusions

Our results highlighted the need to offer psychological support to disaster responders throughout different activity phases. Support during the pre- and during activity periods may be crucial to preventing and minimising the negative impact of stressors associated with disaster response. Promoting self-care in accordance with stress monitoring can play a central role in psychosocial support. Therefore, organisational efforts in routinising self-care by controlling the workload and ensuring self-care opportunities are recommended. Disaster responders often work in an environment where team members and resources from different departments, organisations, and countries are assembled in an ad hoc manner. This condition makes it very difficult to build and operationalise a consistent support system. Co-ordinating disaster-response activities, including workload management and health monitoring of responders, cannot be viewed as dispensable, and the use of external resources may be one of the solutions. Barriers against these recommendations need to be identified in future research to bridge the gap between recommendations and implementation.

## Figures and Tables

**Figure 1 ijerph-17-02011-f001:**
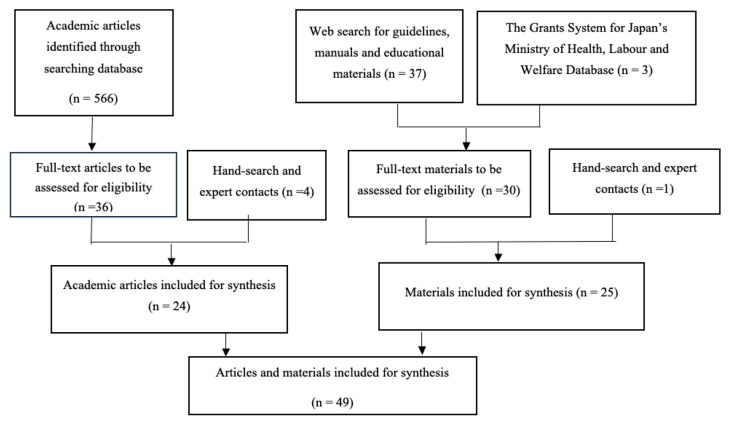
Selection of sources of evidence: academic articles and guidelines, manuals, and educational materials included for synthesis.

**Table 1 ijerph-17-02011-t001:** Actions for Goal 1: Understanding stressors and making them manageable.

Disaster Time	Actors	Actions	References
Before	Individual Actions	1. Gather information on one’s duties and area of operation. 2. Identify possible challenges on site. 3. Assess the readiness of one’s health, work, and family for enrolment. 4. Make an honest decision on whether they could join a disaster-response team.	[27,28,29,30,31,32,33,34,35]
	Organisational Actions	1. Train responders in monitoring their stress levels. 2. Address potential work-related stressors. 3. Consider thoroughly who should or should not be dispatched to disaster-affected areas.	[23,24,27,28,31,32,35,36,37,38,39,40,41,42,43,44,45,46,47]
During	Individual Actions	1. Use a stress checklist to assess the impact of stressors. 2. Accept one’s own emotional reactions and tensions. 3. Re-conceptualise one’s experience during duties, and feelings about them, from different angles.	[29,30,41,48,50]
	Organisational Actions	1. Enable responders to monitor their level of stress. 2. Monitor the physical and mental health of responders.	[25,27,37,38,41,43,45,49,50]
After	Individual Actions	1. Look back on what they experienced and take an objective view of those experiences. 2. Monitor one’s mental health over the long term.	[29,30,35,41,59]
	Organisational Actions	1. Recognise disaster-response activities as a contribution to the missions of organisation. 2. Provide responders with opportunities for frankly talking about their experience and feelings. 3. Monitor responder’s mental health over the long term.	[25,28,35,38,41,43,50,51,52,53,54,55,56,57,58,60]

**Table 2 ijerph-17-02011-t002:** Actions for Goal 2: Reducing stressors and preventing chronically stressful situations.

Disaster Time	Actors	Actions	References
Before	Individual Actions	1. Promote and maintain one’s physical and mental states. 2. Develop one’s own self-care plan. 3. Explain to family members about the duties, and set up support and communication. 4. Disentangle one’s concerns at home and at work.	[28,32,40,41,50]
	Organisational Actions	1. Improve responders’ basic knowledge and skills of stress management. 2. Improve responders’ teamwork skills. 3. Give responders a concrete idea of what their duties will be. 4. Develop means of communication with responders’ family. 5. Develop an efficient operational system with a clear command chain. 6. Have a written and active policy for preventing and managing the stress of responders.	[23,25,27,28,29,32,35,37,38,39,41,44,47,48,50,60,61,70]
During	Individual Actions	1. Maintain routines for one’s health. 2. Get enough rest and refresh oneself using the self-care plan. 3. Refrain from too much alcohol, tobacco, and caffeine. 4. Create mutually supportive teams with co-workers. 5. Keep connected with family and friends. 6. Keep a positive attitude in one’s role.	[24,27,28,29,30,31,32,38,41,42,45,48,50,53,59,63,67,68]
	Organisational Actions	1. Control the volume and content of work given to responders. 2. Hold a defusing meeting to normalise responders’ reactions to stressors. 3. Support responders’ informal communication with their peers. 4. Develop a peer support system within the team (buddy system).	[25,27,28,29,30,37,38,43,44,45,46,48,50,54,60,64,65,66]
After	Individual Actions	1. Switch from a disaster-response to routine mode. 2. Take time off from work to recover from physical and psychological fatigue. 3. Spend time with family and friends.	[27,29,30,59,67,68,69]
	Organisational Actions	1. Clearly announce the end of disaster-response activities. 2. Ensure that responders can take time off work before returning to their routine there. 3. Provide responders with information on self-care.	[24,25,27,28,35,37,39,41,48,50,57,70,72]

**Table 3 ijerph-17-02011-t003:** Actions for Goal 3: Responding to crises and alleviating stress.

Disaster Time	Actors	Actions	References
Before	Individual Actions	Develop personal indicators for extreme fatigue or crisis.	[32]
	Organisational Actions	1. Provide training of psychological first aid to make it immediately available to all responders in times of crisis. 2. Develop a system that responds to the traumatic experiences of responders.	[28,36,37,43,45,47,50,56]
During	Individual Actions	1. Ask for help from mental health professionals. 2. Make a decision as to whether to continue with or resign one’s duties.	[28,30,37,50,55]
	Organisational Actions	1. Give the responder time off, or lighten the volume and content of work. 2. Provide support that is specific to the crisis of that responder. 3. Ensure access to professional support from external organisations. 4. Make a decision as to whether the stressed-out responder should remain at the site or be replaced.	[24,25,28,31,37,42,46,48,56,66]
After	Individual Actions	1. Ask for help from professionals. 2. Take rest until recovery from the mental health crisis.	[37,48,53,56,57,61,71]
	Organisational Actions	Link responders and their families to social resources, including mental health professionals, to provide the responder with mental health care.	[24,25,27,28,30,36,41,48,49,57,59,72]

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
