# Peer review of "A Literature Review on Psychosocial Support for Disaster Responders: Qualitative Synthesis with Recommended Actions for Protecting and Promoting the Mental Health of Responders"

_ijerph, 2020, doi:10.3390/ijerph17062011_

Round 1
Reviewer 1 Report
This paper reviews publications on psychological stress in disaster responders and condenses recommendations for how to alleviate stressors before, during, and after disasters. It is an interesting topic and the paper was fairly well-written and easy to read. However, I think some work needs to be done before publication.
Are you including all the following “health professionals, relief workers, public service providers, and volunteer workers in disaster-affected areas, are at high risk for extreme stress, as are disaster survivors” in the study as disaster responders? This is casting a very wide net—the differences in needs between these groups may be large enough that generalizations are not very helpful. What about firefighters and the like? Do you consider them to be part of this response network? It is not clear from your description whether you are referring only to those who are part of response but lack formal training specific to disasters. Perhaps using a definition of a response network already established by others might help here. Does this review include literature on both natural and human-caused disasters?
While you do note that you purposefully include non-peer-reviewed literature due to lack of empirical evidence to address your question, but it is not clear that this was done systematically. Are all the sources considered equally? Did you have any way to assess whether the source seems “legitimate”? You touch on this in the conclusion, but it would be good to mention it earlier on as well. I am not entirely convinced that the search terms you used were as inclusive as they should have been. For example, you used Google Scholar, but a simple search for “psychological stress in disaster responders” in Google Scholar returns several citations that seem they likely should be cited. Some examples are:
Benedek, D. M., Fullerton, C., & Ursano, R. J. (2007). First responders: mental health consequences of natural and human-made disasters for public health and public safety workers. Annu. Rev. Public Health, 28, 55-68.
Haugen, P. T., Evces, M., & Weiss, D. S. (2012). Treating posttraumatic stress disorder in first responders: A systematic review. Clinical psychology review, 32(5), 370-380.
Hyman, O. (2004). Perceived social support and secondary traumatic stress symptoms in emergency responders. Journal of Traumatic Stress: Official Publication of The International Society for Traumatic Stress Studies, 17(2), 149-156.
This makes your conclusion statement that “further studies are needed to develop an evidence-based package of psychosocial support for disaster responders” ring hollow because it is not clear that the review of the existing literature was systematic or thorough.
In Methods- On page 3 you refer first to post-it notes, then to cards. I assume this refers to the same notes?
While many of the recommendations you condensed in the tables seem sensible, you do not address whether many of them are feasible. For example, taking time off work after disasters is not feasible for many, not least because they have to go back to their regular jobs—at least in the U.S., many disaster responders are part of response teams that are separate from their home units of employment. There also seems to be a lack of understanding of the dominant culture among many disaster responders in a number of the recommendations. Taking time off work “for a considerable amount of time” is among these, as is the use of mental health professionals during this resting phase. Cultural change is important, but you do not acknowledge that such change would be a necessary antecedent for many of the recommendations. You make some mention of potential differences in disaster response roles and contexts by location, but in my opinion do not acknowledge the culture of response—it is like military culture in that it is different from most jobs, and from many professions that provide help or support to others. I think some work is still needed to bridge between the very specific tables and the generalities in the conclusion, and to convince readers that enough literature was reviewed to conclude that we don’t know much about the area.
Author Response
Dear Reviewer 1
We sincerely appreciate your comments and suggestions, and have revised our manuscript accordingly. As one of the reviewers noted, this manuscript can be published as either a Research Report or Communications, whichever is more appropriate. This manuscript was based on the preliminary results of a project that aims to develop a psychosocial support guide for disaster responders that can be used in a global setting. This preliminary study aimed to identify as many psychosocial support items as possible so that they could be scrutinized further for their effectiveness. For this reason, our methodology and presentation may not be academically rigorous. We would greatly appreciate your kind consideration for publication in the most suitable form.
We hope that our revisions have adequately addressed your concerns and made the manuscript suitable for publication. In what follows, we provide our point-by-point responses to your comments along with the corresponding revisions made to the manuscript.
Comments from REVIEWER 1
- Are you including all the following “health professionals, relief workers, public service providers, and volunteer workers in disaster-affected areas, are at high risk for extreme stress, as are disaster survivors” in the study as disaster responders? This is casting a very wide net—the differences in needs between these groups may be large enough that generalizations are not very helpful. What about firefighters and the like? Do you consider them to be part of this response network? It is not clear from your description whether you are referring only to those who are part of response but lack formal training specific to disasters. Perhaps using a definition of a response network already established by others might help here. Does this review include literature on both natural and human-caused disasters?
<Response>
Thank you very much for clarifying this point. Indeed, we did not include the materials for firefighters and self-defense forces in this study because these professional disaster responders and organizations were found to be better prepared for disasters; thus, the support needs of these professionals were quite different from those of other responders. The types of disasters we targeted were natural disasters. However, some materials for which the target was not necessarily a natural disaster, such as Psychological First Aid and the UNHCR Emergency Handbook, provided useful information in regard to psychosocial support for responders in a natural disaster. Considering that our aim was to identify psychosocial support items as thoroughly as possible, we did not exclude these materials.
To address these issues, we added a section to describe the inclusion criteria and explained this as follows:
“The initial focus of this review was psychosocial support for disaster responders who had not received formal training to respond to a natural disaster. Materials developed for firefighters and army personnel were not included because the support needs of these professionals would most likely differ from our target responders owing to the difference in preparedness for critical disaster-related incidents. On the other hand, we did not exclude materials with scopes broader than natural disasters when they provided information applicable to disaster responders.” (Methods, 2.3 Inclusion criteria)
- While you do note that you purposefully include non-peer-reviewed literature due to lack of empirical evidence to address your question, but it is not clear that this was done systematically. Are all the sources considered equally? Did you have any way to assess whether the source seems “legitimate”? You touch on this in the conclusion, but it would be good to mention it earlier on as well. I am not entirely convinced that the search terms you used were as inclusive as they should have been. For example, you used Google Scholar, but a simple
search for “psychological stress in disaster responders” in Google Scholar returns several citations that seem they likely should be cited.
This makes your conclusion statement that “further studies are needed to develop an evidence-based package of psychosocial support for disaster responders” ring hollow because it is not clear that the review of the existing literature was systematic or thorough.
<Response>
We understand your concern that our search may not have been sufficiently thorough. The main purpose of this review was to identify possible types of psychosocial support by referring to field-based knowledge and practices. This is why we reviewed non-academic literature such as guides, manuals, and educational materials. We also reviewed academic articles, but only when they tested the effectiveness of psychosocial support specific to disaster responders as a primary source of evidence. Thus, studies on associated factors and review articles were not reviewed, which is a possible reason for why we did not include some articles.
However, the concern you raised is very reasonable. Therefore, we revised the last paragraph of the Introduction to clarify further the purpose and target of this review as follows.
“The present review was part of a project that aimed to develop a psychosocial support guide for disaster responders that could be used in a global setting. The purpose of this review was to identify types of psychosocial support considered appropriate for disaster responders as a preliminary item pool for guide development. To achieve this aim, we reviewed non-academic literature such as guides, manuals, and educational materials to identify field-based knowledge and practices. Although the information in this review may not be necessarily tested scientifically, it does provide a comprehensive picture of field experiences that can be further scrutinized by scientific measures. Academic articles examining the effectiveness of psychosocial support specific to disaster responders were also reviewed as a primary source of evidence.” (fifth paragraph of the Introduction)
- In Methods- On page 3 you refer first to post-it notes, then to cards. I assume this refers to the same notes?
<Response>
Thank you for pointing this out. We changed “cards” to “post-it notes”.
- While many of the recommendations you condensed in the tables seem sensible, you do not address whether many of them are feasible. For example, taking time off work after disasters is not feasible for many, not least because they have to go back to their regular jobs—at least in the U.S., many disaster responders are part of response teams that are separate from their home units of employment. There also seems to be a lack of understanding of the dominant culture among many disaster responders in a number of the recommendations. Taking time off work “for a considerable amount of time” is among these, as is the use of mental health professionals during this resting phase. Cultural change is important, but you do not acknowledge that such change would be a necessary antecedent for many of the recommendations. You make some mention of potential differences in disaster response roles and contexts by location, but in my opinion do not acknowledge the culture of response—it is like military culture in that it is different from most jobs, and from many professions that provide help or support to others. I think some work is still needed to bridge between the very specific tables and the generalities in the conclusion, and to convince readers that enough literature was reviewed to conclude that we don’t know much about the area.
<Response>
We greatly appreciate your feedback addressing the importance of bridging research findings and field applications. We agree with you regarding the influence of the dominant culture on the work environment and psychosocial well-being of disaster responders. We acknowledge that our manuscript lacked a discussion of these points. Therefore, we added a discussion of the possible barriers against these recommendations and referred to the need to survey the feasibility of these actions as follows:
“However, it should be noted that recommending these actions does not necessary mean that they were implemented in the field. For example, a study conducted in the area affected by the Great East Japan Earthquake found that the largest proportion of firefighters, 42%, answered that long-term leave was the most prioritized measure for treating critical stress-related incidents caused by the disaster response. On the other hand, only 2% answered that their affiliated department offered them a leave after the completion of their response duties. Considering that human resource management for firefighters is often better prepared for disaster situations, the unmet needs of other types of disaster responders, especially volunteer responders, may be even larger.” (seventh paragraph of the Discussion)
“In addition, our review covered recommended actions written in reports, guides, manuals, and research articles. Therefore, how many of these actions were performed in the field, and whether they were effective for protecting and improving the mental health of responders, remains unclear. For example, a systematic review conducted by Guilaran et al. [74] reported that the positive impact of interpersonal support was limited in size to prevent mental health problems among disaster responders. Thus, to develop an evidence-based package of psychosocial support for disaster responders, further studies examining the feasibility and effectiveness of these actions are needed.” (last paragraph of the Discussion)
Reviewer 2 Report
Through the retrieval of relevant materials and databases, the manuscript puts forward actions and specific suggestions for improving and protecting the psychosocial health of disaster responders. The whole research has certain value and it’s interesting, but there are some problems:
The authors need to adjust the format of the manuscript to meet the format requirements of this journal.
In the introduction section, the authors said “over the past two decades, the psychological impact of disasters has come to be one of the major focus of disaster health management”, and I suggest the authors add more literature to justify this view.
In the introduction section, the authors said “Existing studies suggest that programs providing knowledge about stress and stress management could improve the self-esteem of disaster responders, facilitate their self-care, and motivate them to engage in self-directive learning regarding their duties”. Please add more literature to justify this view.
In the introduction section, please add how the sections are organized.
In the method section, the authors said "The procedure of analysis was conducted by three review authors". What institutions do the three authors come from? Are they experienced in the field? Is it too small to select only 3 review authors? The authors need to explain clearly why they were chosen.
In the results section, the authors said “Each action was explained below based on its goal“. The author needs to explain how the three goals are determined? What is the basis?
The discussion section is too weak, and the authors should discuss and make suggestions based on the results of the results chapter analysis.
The conclusion part needs to be further improved. In addition to the need to summarize and analyze the results of this study, it is necessary to make a certain outlook on the future research direction
The paper addresses an important problem. However, the presentation must be improved. It now looks more like a research report than an academic paper.
Author Response
Dear Reviwer 2
We sincerely appreciate your comments and suggestions, and have revised our manuscript accordingly. As you noted, this manuscript can be published as either a Research Report or Communications, whichever is more appropriate. This manuscript was based on the preliminary results of a project that aims to develop a psychosocial support guide for disaster responders that can be used in a global setting. This preliminary study aimed to identify as many psychosocial support items as possible so that they could be scrutinized further for their effectiveness. For this reason, our methodology and presentation may not be academically rigorous. We would greatly appreciate your kind consideration for publication in the most suitable form.
We hope that our revisions have adequately addressed your concerns and made the manuscript suitable for publication. In what follows, we provide our point-by-point responses to your comments along with the corresponding revisions made to the manuscript.
Comments from REVIEWER 2
- The authors need to adjust the format of the manuscript to meet the format requirements of this journal.
<Response>
We apologize for any inconvenience caused by not following the standardized format adequately. We carefully rechecked the format and made corrections accordingly. Please let us know if any inconsistencies remain.
- In the introduction section, the authors said “over the past two decades, the psychological impact of disasters has come to be one of the major focus of disaster health management”, and I suggest the authors add more literature to justify this view.
<Response>
Thank you very much for your suggestion. Accordingly, we added references to this sentence. (Please refer to the second paragraph of Introduction.)
- In the introduction section, the authors said “Existing studies suggest that programs providing knowledge about stress and stress management could improve the self-esteem of disaster responders, facilitate their self-care, and motivate them to engage in self-directive learning regarding their duties”. Please add more literature to justify this view.
<Response>
Based on your suggestion, we added more references to support this claim. (Please refer to the fourth paragraph of the Introduction.)
- In the introduction section, please add how the sections are organized.
<Response>
In accordance with your feedback, we added the following introductory statement at the end of the Introduction:
“In the following sections, we describe the procedure of our review and our analytical framework. Accordingly, our findings are presented using an analytical framework that groups the identified actions by their goals. To enhance their applicability to the field, the identified actions are further grouped by disaster response phase and actors. Lastly, we discuss the characteristics of these actions and challenges in their implementation.”
- In the method section, the authors said "The procedure of analysis was conducted by three review authors". What institutions do the three authors come from? Are they experienced in the field? Is it too small to select only 3 review authors? The authors need to explain clearly why they were chosen.
<Response>
Thank you for your comment. We realized that our description was not accurate. Indeed, all five authors conducted the literature search, and four of them analyzed the data. All authors were from a nursing research institute specializing in disaster health management. Two of the authors are specialized in psychiatric nursing, one in nursing management and disaster nursing, one in acute care management, and the other in public health and epidemiology. All authors were project members for the development of a guide on psychosocial support for disaster responders, and all have field or research experience in disaster-affected areas. We followed the Cochrane Collaboration guidelines, which require a minimum of two persons for every step of a review.
We added “2.1 Review members” as follows to the beginning of the Methods:
“All five authors conducted the literature search, and four of them analyzed the data. All authors were from a nursing research institute specializing in disaster health management. Two of the authors are specialized in psychiatric nursing, one in nursing management and disaster nursing, one in acute care management, and the other in public health and epidemiology. All authors were project members for the development of a guide on psychosocial support for disaster responders, and all have field or research experience in disaster-affected areas.”
- In the results section, the authors said “Each action was explained below based on its goal“. The author needs to explain how the three goals are determined? What is the basis?
<Response>
We agree that the basis of our analytical strategy should have been clearly explained in the manuscript. This manuscript is a part of project that aims to develop a psychosocial support guide for disaster responders that can be used widely in a global context. Because of this aim, we tailored our analytical strategy as follows (please refer to the first paragraph of “2.5 Analysis” in the Methods):
“At the beginning of the analysis, the authors discussed the analytical framework. With the aim of developing a guide applicable to broad societal contexts, the goals of the actions were to enhance field applicability by allowing variations in local contexts. Before setting the goals, we derived some actions from identified materials, and sorted these by disaster response phase. Next, we categorized the actions into a group that aimed at the same goal (these were labeled “Goals”). As follows, we explain in greater detail the procedures in each step by asking research questions.”
- The discussion section is too weak, and the authors should discuss and make suggestions based on the results of the results chapter analysis.
<Response>
In accordance with your feedback, we thoroughly revised the Discussion. We added a summary of our findings across the three goals and added more arguments regarding the types, timings, and actors of identified actions. In response to a comment from Reviewer 1, we also added a new description of the challenges faced in implementing these actions, including the “culture among disaster responders”.
The following paragraphs were added after the first sentence of the Discussion:
“Fifty-five actions to protect and promote the mental health of disaster responders were identified from the reviewed materials, and the following three goals were derived from these actions:; “understand stressors and make them manageable (Goal 1)”, “reduce stressors and prevent chronically stressful situations (Goal 2)”, and “alleviate stressful situations and manage the crisis (Goal 3)”. These three goals can be simply stated in the following three steps: “be aware”, “prevent”, and “respond”.
Among the 55 actions, 17 fell under Goal 1, 25 under Goal 2, and 13 under Goal 3. Therefore, about 70% of the actions were preventive measures (“be aware” and “prevent”). Although responding to a stressor once it occurs is still an important countermeasure against reduced metal health among disaster responders [9], understanding and reducing the occurrence of stressors and enhancing resilience can also be a major part of psychosocial support for disaster responders.
In regard to the timing of actions, most actions were to be performed before and during a disaster response, with 20 actions during a pre-period, 21 during an activity-period, and 14 during a post-period. Specifically, the largest number of actions for Goal 1 (“be aware”) were found in the pre-period, for Goal 3 (“respond”) were in the activity period, and for Goal 2 (“prevent”) in both the pre- and activity periods. These results indicate that many types of psychosocial support were expected to be performed at an earlier stage of a disaster response. This notion echoes the view that pre- and peri-disaster preparation for reducing stressors and responding to stress reactions can help prevent post-disaster mental health problems [5].
Our findings also suggest that individual actions are as important as organizational support for the mental health of disaster responders, as almost the same number of actions were identified for individuals and organizations. Individual actions, such as monitoring one’s own stress level, managing stressors, and performing self-care, were considered to be essential, especially in disaster-affected areas, where each responder is perceived as a caregiver and works under very high-stress conditions with limited human and social resources [4]. Many organizational actions, such as “train responders to monitor their stress” (Goal 1), “improve the responders’ basic knowledge and stress management skills” (Goal 2), and “provide information on self-care” (Goal 2), were undertaken to support such individual actions
Another important role of organizations was to develop a system to monitor, reduce, and respond to responders’ stressors and stress reactions, including “develop an efficient operational system with a clear command chain”, “have a written and active policy for preventing and managing stress among responders”, “control the volume and contents of work”, and “develop a system that responds to the traumatic experiences of responders”. Furthermore, recommendations such as “create mutually supportive teams with other coworkers”, “assure responders of informal communications with their peers”, and “develop a peer support system within the team” suggest that the use of a peer support system may be a promising psychosocial approach to building a resilient response team.”
The following was added to the Discussion (seventh and eighth paragraphs):
“However, it should be noted that recommending these actions does not necessary mean that they were implemented in the field. For example, a study conducted in the area affected by the Great East Japan Earthquake [50] found that the largest proportion of firefighters, 42%, answered that long-term leave was the most prioritized measure for treating critical stress-related incidents caused by the disaster response. On the other hand, only 2% answered that their affiliated department offered them a leave after the completion of their response duties. Considering that human resource management for firefighters is often better prepared for disaster situations, the unmet needs of other types of disaster responders, especially volunteer responders, may be even larger.
One of the possible barriers causing the gap between recommendations and implementation may be conflicts between routine work and a disaster response. In the case that a disaster response is not part of a routine, organizations may not be prepared when their staff leave work behind to respond to a disaster. A lack of supplementary staff to make up for their work would be a heavy burden on their colleagues and management department. Under such conditions, psychosocial support, such as training, monitoring, and offering long-term leave, may not be readily available for responders. In addition, personal characteristics, such as “calm and emotionally collected, acts on logic over emotion, exercises emotional control and self-control”, are highly valued among disaster responders [73]. Therefore, seeking support for one’s mental health conditions can be stigmatized in the culture of responders [50]. Individual counseling for all workers involved may be an effective way of reaching out to such responders [50]. This approach could eventually promote awareness that psychological reactions caused by a disaster response can affect anyone.”
- The conclusion part needs to be further improved. In addition to the need to summarize and analyze the results of this study, it is necessary to make a certain outlook on the future research direction.
<Response>
Based on your feedback, we added key messages to the Discussion and indicated a future research direction. The underlined sentences below were added to the Conclusion:
“Our results highlight the need to offer psychological support to disaster responders throughout different phases of the activities. Pre- and activity periods may be crucial to prevent and minimize the negative impact of stressors associated with a disaster response. Promoting self-care in accordance with stress monitoring can play a central role in psychosocial support. Therefore, organizational efforts in routinizing self-care by controlling workloads and ensuring self-care opportunities are recommended. Disaster responders often work in an environment where team members and resources from different departments, organizations, and countries are assembled in an ad hoc manner, which makes it very difficult to build and operationalize a consistent support system. Coordinating disaster response activities, including the workload management and health monitoring of responders, cannot be viewed as dispensable, and the use of external resources may be one of the solutions. Barriers against these recommendations need to be identified in future research to bridge the gap between recommendations and implementation.”
- The paper addresses an important problem. However, the presentation must be improved. It now looks more like a research report than an academic paper.
<Response>
We are aware that the methodology and presentation of our manuscript may not be academically rigorous. If the editors feel that our manuscript is more suitable as a Research Report than as an academic paper, we defer to their judgment.
Round 2
Reviewer 1 Report
This version of the paper is much improved and was obviously edited carefully.
Author Response
Dear Reviewer 1,
Thank you very much for your time and valuable comments. Owing to your comments, we have been able to improve the quality of our manuscript.
Sincerely,
authors
Reviewer 2 Report
I have no other questions except two small suggestions:
- The authors need to adjust the format of the manuscript.
- The authors need to invite a native speaker to edit the language.
Author Response
Dear Reviewer 2,
Thank you very much for your comments on our manuscript.
Owing to your constructive comments, we have been able to improve the quality of our manuscript. We will further check our English writing, and make necessary correction.
Sincerely,
authors